# At-home wearables and machine learning sensitively capture disease progression in amyotrophic lateral sclerosis

Anoopum S. Gupta [1] ✉, Siddharth Patel [1], Alan Premasiri [2] & Fernando Vieira [2]

Amyotrophic lateral sclerosis causes degeneration of motor neurons, resulting in progressive muscle weakness and impairment in motor function. Promising drug development efforts have accelerated in amyotrophic lateral sclerosis, but are constrained by a lack of objective, sensitive, and accessible outcome measures. Here we investigate the use of wearable sensors, worn on four limbs at home during natural behavior, to quantify motor function and disease progression in 376 individuals with amyotrophic lateral sclerosis. We use an analysis approach that automatically detects and characterizes submovements from passively collected accelerometer data and produces a machine-learned severity score for each limb that is independent of clinical ratings. We show that this approach produces scores that progress faster than the gold standard Amyotrophic Lateral Sclerosis Functional Rating Scale-Revised ($-0.86 \pm 0.70$ SD/year versus $-0.73 \pm 0.74$ SD/year), resulting in smaller clinical trial sample size estimates ($N = 76$ versus $N = 121$). This method offers an ecologically valid and scalable measure for potential use in amyotrophic lateral sclerosis trials and clinical care.

Novel therapeutic modalities are now aimed at proximal disease mechanisms in amyotrophic lateral sclerosis (ALS) and other neurodegenerative diseases[1,2]. One major barrier to the successful and efficient development of disease-modifying therapies for neurodegenerative disorders is a lack of objective clinical outcome measures that account for disease heterogeneity and can sensitively quantify disease progression over the duration of a clinical trial[3–5]. The standard tool for assessing disease severity in ALS clinical trials and clinical care is a semi-quantitative rating scale (ALS Functional Rating Scale-Revised[6,7] or ALSFRS-R) that uses multiple choice questions to evaluate several behavioral functions (e.g., walking, handwriting, speech, swallowing). The assessment is most often completed by clinicians specializing in ALS[7,8], however recent studies have shown high correlation between clinician-performed ALSFRS-R and at-home, patient-performed ALSFRS-R[9]. Clinician or patient-performed ALSFRS-R is a useful assessment of global motor function; however,

it is subjective, categorical, and is only performed intermittently over time, which limits its sensitivity for measuring disease change and contributes to the need for relatively large and expensive trials[10,11]. This is a particular challenge in rare diseases and results in pressure to include relatively homogenous cohorts with faster rates of disease progression, which restricts participation of some individuals and may not be representative of the entire ALS population[12].

There is a great opportunity to reduce the size and cost of ALS trials, increase the population of individuals who can participate, and accelerate the evaluation of promising therapeutics through the development of new categories of sensitive quantitative motor outcome measures[13–15]. Quantitative motor outcome measures may be task-based (i.e., measuring behavior during the performance of a specific task) or task-free, where an individual's natural behavior is measured passively and continuously at home. There has been recent development of several task-based approaches to quantify speech and limb function in ALS

[1]Department of Neurology, Massachusetts General Hospital, Harvard Medical School, Boston, MA, USA. [2]ALS Therapy Development Institute, Watertown, MA, USA. ✉e-mail: agupta@mgh.harvard.edu

using scalable technologies at home[9,16–18] and only a single report of a task-free approach in ALS using a waist-worn accelerometer[19]. Task-based measures, however, have some of the same limitations as rating scales in that they are based on a relatively small number of data samples and cannot easily account for diurnal and day-to-day variability, they rely on the participant's ability and motivation to perform the task, and they are susceptible to learning and placebo effects.

Task-free assessment approaches which passively and continuously measure natural behavior at home have the potential to overcome these limitations and be transformative by making reliable and sensitive measures available at scale. Furthermore, they have the potential to produce measures that more closely reflect the day-to-day function of the individual by measuring the individual's own selection of behaviors. However, the information obtained by the tool must be interpretable and meaningful to support its use in clinical trials or clinical care.

Here we demonstrate that a submovement-focused analysis of triaxial accelerometer data[20,21], recorded from wrist and ankle sensors worn by hundreds of individuals with ALS at home during natural behavior, produces interpretable and robust measures of motor function and disease progression. We develop a machine learning approach to train a model that is sensitive to disease change by utilizing the information for how individuals' sensor-based movement patterns change over time, rather than being constrained by existing clinical assessments such as ALSFRS-R. We show that the model's severity estimates and longitudinal trajectories are reliable and consistent with ALSFRS-R, but are more sensitive than the clinical scale for measuring change over time. Thus, we demonstrate that objective, sensitive, and scalable measures of motor function and disease change can be obtained from passive analysis of everyday behavior using inexpensive wearable sensors.

## Results

### Overview of the dataset

We analyzed accelerometer data from wrist and ankle-worn sensors collected as part of the Precision Medicine Program launched by the ALS Therapy Development Institute (ALS TDI) in 2014 (see Methods). Individuals were asked to wear a sensor on each wrist and ankle continuously for a week each month. Participants also performed a sequence of 5 limb-based exercises on alternating days, lasting a total of approximately 5 min. Participants were instructed that sensors must be worn during the brief exercises, but to also wear the sensors as much as possible throughout the week without further specifying periods of wear time. An analysis of accelerometer data collected only during the task-based assessments was previously reported[17]. Here, we analyze the entirety of accelerometer data collected at home as individuals performed their typical daily routine without any constraints. Participant clinical and demographic data are shown in Fig. 1A, including the median ALSFRS-R at the study start and study end. 95% of participants lived in the United States (41 states represented), 3% lived in Canada, and 1% lived in the United Kingdom. 93.5% of participants were White, 2% Hispanic, 2% Asian, 1% Black, <1% Middle Eastern, and <1% Polish. 15% of ALS participants had a family history of ALS. The dataset filtering steps are described in Fig. 1B. Cross-sectional analysis included 4637 sessions from 402 unique participants (376 ALS, 26 controls) with at least 24 h of recorded accelerometer data, pooled only from days with at least 3 h of data, from all four limbs (Fig. 1B). The 24 h session minimum for daytime data was chosen based on prior work demonstrating high reliability of daytime data across the first three and last three days in a week[20,21]. Longitudinal analysis was conducted using data from participants with at least three data collection sessions spanning a minimum of 0.75 years (188 ALS and 6 control participants). Participants had a median of 9 days per session with at least 3 h/day of data and averaged 8.9 h/day of wear time for the wrist sensors and 7.4 h/day for the ankle sensors. The duration of daily sensor wear time (averaged across the four sensors) decreased over

the course of the study from an average of 9.1 h/day (first session) to 6.5 h/day (last session). To understand the burden of wearing the four sensors periodically over a 0.75-year period (relative to at-home self-report of ALSFRS-R), we identified the subset of the 402 individuals with adequate cross-sectional data who did not wear the sensors for at least 0.75 years but continued to perform ALSFRS-R self-report for 90 days or more after the last time they wore sensors. This consisted of 39 participants or ~10% of the cohort who continued performing ALSFRS-R but stopped wearing the sensors.

Submovement, activity bout, activity index, and spectral movement features (85 total) were extracted from each session as previously described[20,21] (Fig. 1C, Supplementary Table 1). Briefly, continuous triaxial accelerometer data was processed to identify activity bouts (short periods of continuous movement), which were projected onto a 2D plane using principal component analysis to identify the primary and secondary directions of motion[20,21]. The acceleration time series was converted to a velocity time series via integration. Submovements (i.e., typically bell-shaped velocity-time curves flanked by zero velocity crossings, see Fig. 1C) were then identified in the primary and secondary directions of motion and grouped into long and short-duration submovements. Single feature analysis was performed on a subset of 24 key submovement (SM) features of interest. These included SM distance, peak velocity, and peak acceleration (8 features each). Mean and standard deviation were computed for short-duration and long-duration SMs in the primary and secondary directions of planar movement resulting in 8 features for each measurement type.

### Overview of the pairwise comparisons severity estimation model

The task was to train a machine learning model that could combine information across the 85 movement features, previously shown to strongly reflect motor function in pediatric and adult ataxias[20,21], to produce an ALS-specific composite measure that was sensitive to disease progression. The standard machine learning approach is to train a regression model to predict the clinical scale score (e.g., ALSFRS-R). However, the sensitivity of the model is then constrained by the sensitivity of the scale. In the "pairwise model" approach, the model is trained to learn the steepest direction of disease change in feature space based on longitudinal data, without using clinician or patient-reported information. This approach, described in Fig. 2, can be applied to any disease that progresses over time. The model produces a score in which lower values represent increased impairment (as in ALSFRS-R) and there is no lower or upper bound on the value of the score, although scores in the current population ranged from −11.3 to 9.6. In addition to the pairwise model, linear regression models with L1-regularization[22] were trained to predict ALSFRS-R total, ALSFRS-R gross motor subscore, and ALSFRS-R fine motor subscore, and were evaluated using five-fold cross-validation.

### Cross-sectional properties of ankle and wrist sensor data

Individual right ankle SM features, including SM distance, velocity, and acceleration were significantly correlated with ALSFRS-R total ($r = 0.31-0.58$), demonstrated high test−retest reliability (ICC = 0.71–0.93), and were significantly different between ALS and control participants (Table 1). All submovement features were positively correlated with ALSFRS-R, indicating that submovement distances, peak velocities, and peak accelerations were smaller and less variable in individuals with more severe disease. Similarly, right wrist submovement (SM) features were positively correlated with ALSFRS-R total ($r = 0.31-0.48$) and were significantly different between ALS and control participants (Table 2). Long-duration wrist submovements showed high test−retest reliability (ICC = 0.86–0.91), whereas short-duration submovements had moderate test−retest reliability (ICC = 0.55–0.83). Ankle submovement features were more strongly correlated with the ALSFRS-R gross motor subscore ($r = 0.40-0.68$) than with the ALSFRS-R fine motor subscore ($r = 0.16-0.51$) and were only weakly correlated

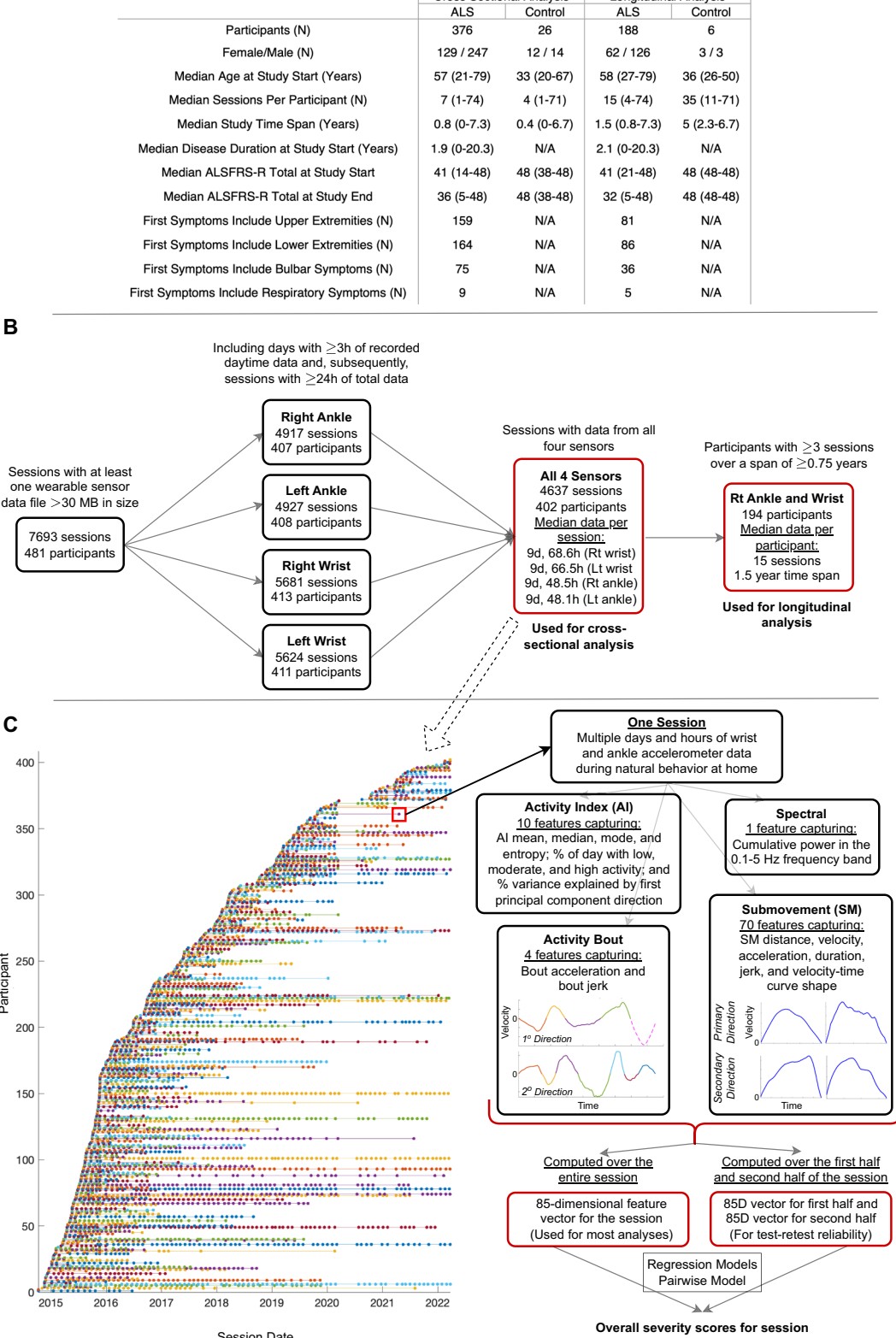

**Fig. 1 | Overview of population and dataset. A** Participant clinical and demographic data with range of values provided in parentheses. **B** Filtering steps for inclusion of sessions and participants used in cross-sectional and longitudinal analysis. **C** Visualization of each participant's session time points for data collection from 2014 to 2022, along with the movement features extracted from each session. ALSFRS ALS Functional Rating Scale-Revised, d day, h hour, Lt left, Rt right.

with respiratory and bulbar subscores. Conversely, wrist submovement features correlated more strongly with the ALSFRS-R fine motor subscore ($r = 0.40$–$0.60$) compared with ALSFRS-R gross motor subscore ($r = 0.19$–$0.32$), and also only weakly correlated with the respiratory and bulbar subscores. Both ankle and wrist submovements demonstrated good agreement between right and left limbs, however ankle right/left agreement ($r = 0.81$–$0.97$) was stronger than wrist right/left agreement ($r = 0.65$–$0.82$).

## A
### Pairwise Comparisons

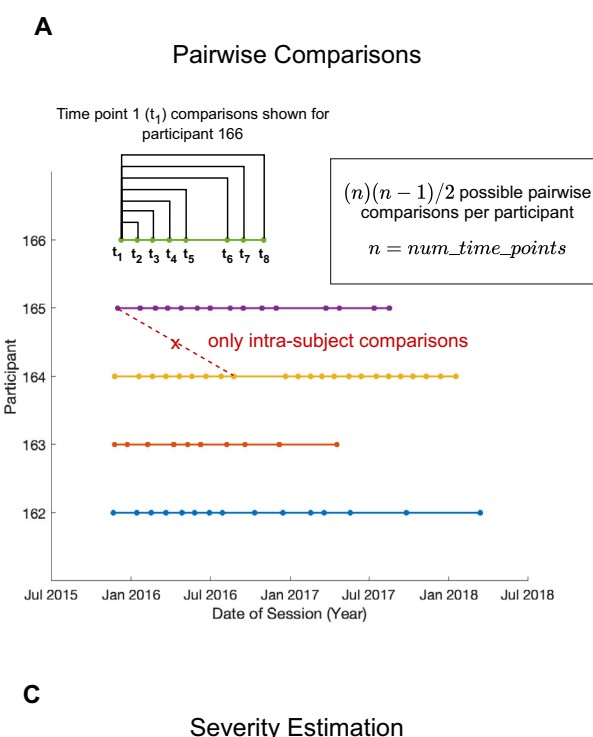

## B
### Classification Model

Train model to predict whether sample 1 (S1) comes before or after sample 2 (S2) in time, based on difference vector. In essence, the model learns the direction in feature space that represents disease worsening (forward in time) versus disease improvement (backward in time).

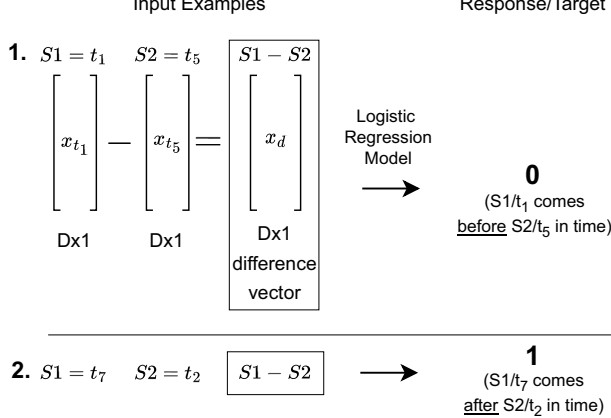

## C
### Severity Estimation

1. Train logistic regression model based on difference vectors as described above:

$$p(x_d) = \frac{1}{1 + e^{-x_d \beta}}$$

2. Apply learned model weights ($\beta$) to original feature vectors to generate a severity score at each time point:

$$severity(x_t) = x_t \beta$$

**Fig. 2 | Overview of the pairwise comparisons model. A** Schematic showing the intra-subject pairwise session comparisons performed for participant 166's first session. For each individual there are $n^*(n-1)/2$ possible pairwise comparisons (where $n$ represents the number of time points or sessions for that individual). **B** The model takes two 85-dimensional feature vectors or samples ($S_1$ and $S_2$) from a single individual as input, representing that individual's motor function at two different points in time ($t_m$ and $t_n$). For each comparison, the samples at times $t_m$ and $t_n$ are randomly assigned to be $S_1$ and $S_2$, such that for approximately half the comparisons $S_1$ is the earlier time sample (as shown in B.1.) and for the other half $S_1$ is the later time sample (B.2.). The element-wise difference between the two vectors is computed ($S_1$-$S_2$), representing the direction of change in feature space. This difference vector is the input to a binary classifier (logistic regression) which learns to predict whether the direction of change reflects disease progression ($S_2$ is temporally after $S_1$) or reflects disease improvement ($S_2$ is temporally before $S_1$). **C** The learned logistic regression model parameters (representing the direction of disease progression) were then applied as linear weights to the original feature vectors to generate a score that reflects how far in the direction of disease progression the individual had traveled at that moment in time.

Machine learning models trained to learn a composite severity score based on right ankle movement features, correlated well with ALSFRS-R total and ALSFRS-R gross motor subscore ($r = 0.66$-$0.77$), had high test–retest reliability (ICC = 0.88–0.92), distinguished between ALS participants and controls, and demonstrated strong right/left limb agreement ($r = 0.91$-$0.95$, see Table 1). For the right wrist, composite severity scores correlated well with ALSFRS-R total and ALSFRS-R fine motor subscore ($r = 0.65$-$0.72$), had high test–retest reliability (ICC = 0.84–0.90), distinguished between ALS participants and controls, and demonstrated strong right/left limb agreement ($r = 0.82$-$0.86$, see Table 2). Composite scores correlated more strongly with ALSFRS-R for male participants compared to female participants, however test–retest reliability and right/left limb agreement was similar for both groups (Supplementary Tables 2–5). The ankle and wrist pairwise models had the highest test–retest reliability among the machine learning models and were the focus of longitudinal analysis. To understand which individual features were the most salient in the ankle and wrist pairwise models, we identified

features that were in the top five (out of 85) in feature importance for all 5 cross-validation folds. For the right ankle pairwise model, the features included SM peak velocity (mean, PC2 direction, long duration SM group) and SM distance (mean, PC2 direction, long duration SMs). For the right wrist pairwise model the most salient features were SM peak velocity (mean, PC2 direction, long duration SMs) and SM peak velocity (mean, PC2 direction, short duration SMs). Although SM peak velocity was strongly represented in the models, the properties of peak velocity at an individual feature level (e.g., relationships with ALSFRS-R, test–retest reliability) were comparable to SM acceleration and distance features and showed weaker relationships with ALSFRS-R compared to the pairwise models (see Tables 1, 2).

### Longitudinal properties of ankle and wrist sensor data

The rate of change over time for each sensor-based composite score and ALSFRS-R score was modeled using linear regression, with the slope of the best fit line determining the rate of change[23]. To compare the rate of change of different scores, each with a different range of

**Table 1 | Cross-sectional properties of ankle submovement models and features**

| Feature name | Stat. | SM dur. group | Dir. of motion group | Relationship with ALSFRS-R | | | | | | | | | | Test re-test rel. | ALS vs Control | | Right and left ankle agreement | |
|---|---|---|---|---|---|---|---|---|---|---|---|---|---|---|---|---|---|---|
| | | | | Total | | Gross motor | | Fine motor | | Respiratory | | Bulbar | | | | | | |
| | | | | r | p-val | r | p-val | r | p-val | r | p-val | r | p-val | ICC | p-val | es | r | p-val |
| ALSFRS total prediction model | | | | 0.59 | 0 | 0.66 | 0 | 0.52 | 6.0E−293 | 0.26 | 2.0E−66 | 0.20 | 4.0E−41 | 0.88 | 2.0E−56 | 1.1 | 0.91 | 0 |
| ALSFRS Gross Motor Prediction Model | | | | 0.54 | 0 | 0.77 | 0 | 0.40 | 3.0E−162 | 0.21 | 7.0E−43 | 0.11 | 7.0E−12 | 0.91 | 2.0E−78 | 1.4 | 0.95 | 0 |
| Pairwise Model | | | | 0.56 | 0 | 0.75 | 0 | 0.43 | 2.0E−191 | 0.22 | 6.0E−49 | 0.16 | 4.0E−25 | 0.92 | 3.0E−73 | 1.3 | 0.94 | 0 |
| SM Dist. | M | Long | PC1 | 0.36 | 9.0E−134 | 0.45 | 3.0E−212 | 0.23 | 7.0E−53 | 0.26 | 2.0E−68 | 0.09 | 9.0E−10 | 0.85 | 5.0E−48 | 1.0 | 0.90 | 0 |
| | M | Long | PC2 | 0.48 | 2.0E−249 | 0.56 | 0 | 0.37 | 3.0E−144 | 0.30 | 2.0E−90 | 0.12 | 4.0E−15 | 0.90 | 8.0E−57 | 1.1 | 0.94 | 0 |
| | M | Short | PC1 | 0.50 | 5.0E−269 | 0.56 | 0 | 0.46 | 3.0E−226 | 0.22 | 4.0E−47 | 0.13 | 7.0E−19 | 0.90 | 7.0E−42 | 0.9 | 0.92 | 0 |
| | M | Short | PC2 | 0.54 | 0 | 0.60 | 0 | 0.48 | 9.0E−253 | 0.27 | 4.0E−70 | 0.15 | 2.0E−21 | 0.91 | 9.0E−42 | 0.9 | 0.93 | 0 |
| | SD | Long | PC1 | 0.31 | 3.0E−96 | 0.40 | 7.0E−163 | 0.16 | 5.0E−27 | 0.23 | 7.0E−54 | 0.10 | 4.0E−10 | 0.71 | 3.0E−49 | 0.9 | 0.81 | 0 |
| | SD | Long | PC2 | 0.39 | 5.0E−154 | 0.46 | 4.0E−228 | 0.26 | 4.0E−67 | 0.26 | 2.0E−66 | 0.12 | 7.0E−15 | 0.75 | 2.0E−59 | 1.1 | 0.86 | 0 |
| | SD | Short | PC1 | 0.52 | 2.0E−305 | 0.63 | 0 | 0.46 | 5.0E−227 | 0.23 | 6.0E−51 | 0.14 | 7.0E−19 | 0.90 | 8.0E−64 | 1.2 | 0.91 | 0 |
| | SD | Short | PC2 | 0.54 | 0 | 0.66 | 0 | 0.47 | 1.0E−231 | 0.24 | 6.0E−59 | 0.14 | 4.0E−21 | 0.88 | 6.0E−71 | 1.3 | 0.93 | 0 |
| SM Vel. | M | Long | PC1 | 0.49 | 2.0E−264 | 0.56 | 0 | 0.37 | 3.0E−142 | 0.30 | 2.0E−90 | 0.16 | 5.0E−26 | 0.88 | 7.0E−56 | 1.1 | 0.92 | 0 |
| | M | Long | PC2 | 0.57 | 0 | 0.65 | 0 | 0.47 | 8.0E−234 | 0.31 | 2.0E−95 | 0.16 | 3.0E−26 | 0.92 | 4.0E−63 | 1.2 | 0.96 | 0 |
| | M | Short | PC1 | 0.50 | 7.0E−277 | 0.59 | 0 | 0.46 | 2.0E−225 | 0.21 | 2.0E−42 | 0.14 | 3.0E−19 | 0.91 | 2.0E−38 | 0.8 | 0.92 | 0 |
| | M | Short | PC2 | 0.54 | 0 | 0.63 | 0 | 0.48 | 5.0E−248 | 0.25 | 8.0E−65 | 0.15 | 6.0E−22 | 0.91 | 3.0E−38 | 0.9 | 0.93 | 0 |
| | SD | Long | PC1 | 0.37 | 7.0E−137 | 0.44 | 2.0E−202 | 0.22 | 5.0E−50 | 0.23 | 2.0E−54 | 0.14 | 1.0E−20 | 0.71 | 4.0E−55 | 1.0 | 0.83 | 0 |
| | SD | Long | PC2 | 0.49 | 2.0E−262 | 0.59 | 0 | 0.36 | 2.0E−132 | 0.26 | 6.0E−67 | 0.17 | 3.0E−28 | 0.80 | 3.0E−77 | 1.3 | 0.91 | 0 |
| | SD | Short | PC1 | 0.52 | 8.0E−294 | 0.63 | 0 | 0.45 | 7.0E−219 | 0.21 | 3.0E−45 | 0.13 | 3.0E−17 | 0.91 | 8.0E−52 | 1.1 | 0.91 | 0 |
| | SD | Short | PC2 | 0.55 | 0 | 0.68 | 0 | 0.47 | 6.0E−237 | 0.24 | 5.0E−57 | 0.14 | 3.0E−20 | 0.90 | 2.0E−59 | 1.1 | 0.92 | 0 |
| SM Accel. | M | Long | PC1 | 0.58 | 0 | 0.63 | 0 | 0.51 | 2.0E−285 | 0.26 | 4.0E−69 | 0.20 | 3.0E−37 | 0.93 | 5.0E−48 | 1.0 | 0.95 | 0 |
| | M | Long | PC2 | 0.58 | 0 | 0.66 | 0 | 0.51 | 2.0E−284 | 0.27 | 1.0E−73 | 0.17 | 3.0E−30 | 0.93 | 2.0E−50 | 1.0 | 0.97 | 0 |
| | M | Short | PC1 | 0.49 | 2.0E−262 | 0.59 | 0 | 0.44 | 3.0E−209 | 0.19 | 7.0E−38 | 0.13 | 7.0E−18 | 0.92 | 5.0E−37 | 0.8 | 0.91 | 0 |
| | M | Short | PC2 | 0.53 | 0 | 0.63 | 0 | 0.47 | 2.0E−233 | 0.24 | 6.0E−58 | 0.14 | 3.0E−20 | 0.91 | 2.0E−37 | 0.8 | 0.92 | 0 |
| | SD | Long | PC1 | 0.55 | 0 | 0.63 | 0 | 0.48 | 6.0E−253 | 0.21 | 8.0E−46 | 0.18 | 4.0E−32 | 0.92 | 2.0E−52 | 1.0 | 0.92 | 0 |
| | SD | Long | PC2 | 0.57 | 0 | 0.68 | 0 | 0.48 | 3.0E−253 | 0.23 | 2.0E−54 | 0.17 | 4.0E−27 | 0.92 | 8.0E−56 | 1.1 | 0.94 | 0 |
| | SD | Short | PC1 | 0.49 | 2.0E−264 | 0.62 | 0 | 0.43 | 3.0E−193 | 0.19 | 3.0E−37 | 0.12 | 2.0E−15 | 0.92 | 3.0E−42 | 0.9 | 0.89 | 0 |
| | SD | Short | PC2 | 0.54 | 0 | 0.67 | 0 | 0.46 | 6.0E−222 | 0.22 | 3.0E−50 | 0.13 | 2.0E−18 | 0.91 | 2.0E−48 | 1.0 | 0.91 | 0 |

For relationships with ALSFRS-R and right-left agreement, p-values for Pearson's correlation were computed using a Student's t distribution for a transformation of the correlation (two-tailed test). Mann–Whitney U test was used for ALS versus control comparisons (two-sided test) and Cohen's d was used to measure effect size. The Benjamini–Hochberg method was used to adjust for multiple comparisons and corrected p-values are reported.

Stat statistic, Dur duration, Dir direction, Dist distance, Vel velocity, Accel acceleration, SM submovement, M mean, SD standard deviation, AI activity intensity, PC principal component, ALSFRS-R ALS Functional Rating Scale-Revised, ICC intraclass correlation coefficient, r Pearson correlation coefficient, es effect size.

values, each score was standardized (subtracting the mean and dividing by the standard deviation) and expressed as a z-score. Rate of change for each score was reported as z-score per year or equivalently as standard deviations (SD) per year.

The rate of change of the pairwise model composite score was computed for each limb. Rate of change was highly consistent across right and left ankles ($r = 0.87$) and right and left wrists ($r = 0.80$, Fig. 3A). There was lesser agreement ($r = 0.52$–$0.56$) between each upper and lower limb pair (e.g., right ankle versus right wrist). Individual-level trajectories demonstrated examples in which all four limbs progressed similarly over time (Fig. 3B), the lower limb pair had similar trajectories but differed from the upper limbs (Fig. 3C), and where the trajectory of one or two limbs deviated from the others (Fig. 3D).

There was also congruence between lower limb pairwise model trajectories and ALSFRS-R gross motor subscore trajectories and between upper limb and ALSFRS-R fine motor subscore trajectories (Fig. 3B–D). The population-level agreement between the right ankle pairwise model rate of change and ALSFRS-R gross motor rate of change ($r = 0.73$, $p = 1.5 \times 10^{-33}$) was stronger than the agreement with ALSFRS-R fine motor ($r = 0.56$, $p = 1.4 \times 10^{-17}$), and the right wrist pairwise model rate of change showed stronger agreement with ALSFRS-R fine motor ($r = 0.73$, $p = 1.1 \times 10^{-33}$) compared to ALSFRS-R gross motor rate of change ($r = 0.60$, $p = 4.2 \times 10^{-20}$).

Next, for each participant, the pairwise model rate of change was combined over the four limbs by either taking the average rate of change or the maximum rate of change. When taking the average of the four limbs, the pairwise model rate of change had strong agreement with ALSFRS-R total rate of change ($r = 0.71$), gross motor subscore rate of change ($r = 0.75$), and fine motor subscore rate of change ($r = 0.68$, Fig. 4A), and weak agreement with respiratory and bulbar subscores ($r = 0.38$ and $r = 0.45$, respectively). Similarly, when taking the limb with the maximum rate of change, the pairwise model had strong agreement with ALSFRS-R ($r = 0.69$), gross motor subscore ($r = 0.75$), and fine motor subscore ($r = 0.69$, Fig. 4B), and weak agreement with respiratory and bulbar subscores ($r = 0.34$ and $r = 0.43$, respectively). The sensor-based pairwise model, which was trained to estimate disease severity without knowledge of ALSFRS-R scores, had strong rate-of-change agreement with the regression model trained to estimate ALSFRS-R total score, regardless of whether the average of the four limbs or the limb with the fastest progression rate was used ($r = 0.92$ for both, Fig. 4A, B). Thus, averaging or taking the maximum rate of change across the four limbs produced equally robust and consistent measures of disease progression.

When taking the maximum rate of change, points shift downward with respect to the $y = x$ line (Fig. 4A versus 4B) indicating increased sensitivity of the sensor-based model to disease change in comparison with ALSFRS-R total. Using the maximum rate of change, the pairwise model had a progression rate of $-0.86 \pm 0.70$ (mean ± standard deviation) SD/year and the regression model had a progression rate of $-0.86 \pm 0.74$ SD/year. Both the pairwise and regression models

**Table 2 | Cross-sectional properties of wrist submovement models and features**

| Feature name | Stat. | SM dur. group | Dir. of motion group | Relationship with ALSFRS-R | | | | | | | | | | Test re-test rel. | ALS vs Control | | Right and left wrist agreement | |
|---|---|---|---|---|---|---|---|---|---|---|---|---|---|---|---|---|---|---|
| | | | | Total | | Gross motor | | Fine motor | | Respiratory | | Bulbar | | | | | | |
| | | | | r | p-val | r | p-val | r | p-val | r | p-val | r | p-val | ICC | p-val | es | r | p-val |
| ALSFRS Total Prediction Model | | | | 0.62 | 0 | 0.53 | 0 | 0.65 | 0 | 0.28 | 5.0E-77 | 0.27 | 2.0E-72 | 0.84 | 7.0E-88 | 1.4 | 0.86 | 0 |
| ALSFRS Fine Motor Prediction Model | | | | 0.61 | 0 | 0.46 | 3.0E-224 | 0.72 | 0 | 0.27 | 2.0E-71 | 0.25 | 3.0E-64 | 0.85 | 1.0E-87 | 1.4 | 0.82 | 0 |
| Pairwise Model | | | | 0.58 | 0 | 0.43 | 2.0E-196 | 0.65 | 0 | 0.21 | 4.0E-44 | 0.30 | 5.0E-89 | 0.90 | 7.0E-114 | 1.7 | 0.86 | 0 |
| SM Dist. | M | Long | PC1 | 0.38 | 5.0E-146 | 0.24 | 6.0E-59 | 0.52 | 9.0E-303 | 0.17 | 3.0E-30 | 0.10 | 6.0E-11 | 0.86 | 6.0E-19 | 0.6 | 0.66 | 0 |
| | M | Long | PC2 | 0.38 | 4.0E-150 | 0.24 | 5.0E-59 | 0.51 | 3.0E-290 | 0.19 | 2.0E-37 | 0.11 | 2.0E-12 | 0.86 | 3.0E-29 | 0.7 | 0.72 | 0 |
| | M | Short | PC1 | 0.37 | 7.0E-144 | 0.22 | 8.0E-49 | 0.49 | 3.0E-260 | 0.11 | 8.0E-14 | 0.20 | 2.0E-38 | 0.74 | 3.0E-50 | 0.9 | 0.72 | 0 |
| | M | Short | PC2 | 0.41 | 3.0E-175 | 0.25 | 2.0E-64 | 0.54 | 0 | 0.15 | 7.0E-22 | 0.19 | 7.0E-35 | 0.83 | 3.0E-47 | 1.0 | 0.76 | 0 |
| | SD | Long | PC1 | 0.45 | 2.0E-217 | 0.30 | 2.0E-90 | 0.58 | 0 | 0.20 | 1.0E-37 | 0.17 | 2.0E-29 | 0.88 | 2.0E-28 | 0.8 | 0.69 | 0 |
| | SD | Long | PC2 | 0.46 | 6.0E-221 | 0.29 | 5.0E-84 | 0.58 | 0 | 0.21 | 9.0E-45 | 0.18 | 2.0E-33 | 0.88 | 8.0E-45 | 1.0 | 0.73 | 0 |
| | SD | Short | PC1 | 0.35 | 2.0E-128 | 0.21 | 7.0E-45 | 0.44 | 3.0E-209 | 0.10 | 2.0E-10 | 0.21 | 7.0E-45 | 0.55 | 7.0E-63 | 0.8 | 0.65 | 0 |
| | SD | Short | PC2 | 0.42 | 7.0E-184 | 0.26 | 8.0E-67 | 0.54 | 0 | 0.12 | 4.0E-16 | 0.22 | 5.0E-49 | 0.73 | 2.0E-70 | 1.1 | 0.72 | 0 |
| SM Vel. | M | Long | PC1 | 0.40 | 2.0E-163 | 0.25 | 3.0E-63 | 0.54 | 0 | 0.18 | 7.0E-31 | 0.13 | 4.0E-18 | 0.88 | 1.0E-25 | 0.7 | 0.68 | 0 |
| | M | Long | PC2 | 0.40 | 9.0E-169 | 0.25 | 6.0E-65 | 0.53 | 0 | 0.19 | 5.0E-37 | 0.14 | 2.0E-20 | 0.88 | 4.0E-38 | 0.9 | 0.73 | 0 |
| | M | Short | PC1 | 0.39 | 5.0E-154 | 0.22 | 5.0E-49 | 0.50 | 2.0E-268 | 0.12 | 6.0E-15 | 0.22 | 3.0E-47 | 0.75 | 2.0E-63 | 1.1 | 0.74 | 0 |
| | M | Short | PC2 | 0.42 | 3.0E-180 | 0.26 | 1.0E-65 | 0.53 | 0 | 0.15 | 7.0E-22 | 0.20 | 5.0E-41 | 0.81 | 2.0E-54 | 1.1 | 0.77 | 0 |
| | SD | Long | PC1 | 0.48 | 1.0E-249 | 0.32 | 3.0E-100 | 0.60 | 0 | 0.20 | 4.0E-39 | 0.21 | 3.0E-45 | 0.90 | 7.0E-40 | 0.9 | 0.71 | 0 |
| | SD | Long | PC2 | 0.48 | 2.0E-248 | 0.31 | 3.0E-93 | 0.59 | 0 | 0.21 | 7.0E-43 | 0.23 | 6.0E-51 | 0.90 | 5.0E-63 | 1.2 | 0.75 | 0 |
| | SD | Short | PC1 | 0.35 | 2.0E-125 | 0.21 | 2.0E-43 | 0.44 | 3.0E-204 | 0.09 | 7.0E-10 | 0.21 | 6.0E-45 | 0.52 | 2.0E-77 | 0.8 | 0.67 | 0 |
| | SD | Short | PC2 | 0.42 | 7.0E-181 | 0.26 | 5.0E-67 | 0.53 | 0 | 0.12 | 7.0E-16 | 0.22 | 2.0E-49 | 0.68 | 2.0E-78 | 1.1 | 0.72 | 0 |
| SM Accel. | M | Long | PC1 | 0.43 | 2.0E-189 | 0.25 | 1.0E-63 | 0.55 | 0 | 0.15 | 6.0E-24 | 0.21 | 7.0E-44 | 0.91 | 2.0E-52 | 1.1 | 0.76 | 0 |
| | M | Long | PC2 | 0.43 | 6.0E-190 | 0.26 | 5.0E-66 | 0.54 | 0 | 0.17 | 3.0E-28 | 0.21 | 2.0E-43 | 0.90 | 6.0E-58 | 1.1 | 0.78 | 0 |
| | M | Short | PC1 | 0.38 | 2.0E-145 | 0.21 | 9.0E-45 | 0.48 | 7.0E-248 | 0.11 | 8.0E-14 | 0.22 | 5.0E-49 | 0.72 | 9.0E-70 | 1.1 | 0.73 | 0 |
| | M | Short | PC2 | 0.41 | 2.0E-171 | 0.25 | 2.0E-62 | 0.51 | 4.0E-291 | 0.14 | 7.0E-20 | 0.21 | 2.0E-43 | 0.78 | 9.0E-58 | 1.1 | 0.78 | 0 |
| | SD | Long | PC1 | 0.46 | 4.0E-229 | 0.29 | 3.0E-84 | 0.55 | 0 | 0.15 | 2.0E-21 | 0.28 | 7.0E-79 | 0.90 | 6.0E-74 | 1.3 | 0.81 | 0 |
| | SD | Long | PC2 | 0.45 | 6.0E-218 | 0.29 | 1.0E-81 | 0.54 | 0 | 0.14 | 2.0E-21 | 0.27 | 7.0E-72 | 0.89 | 6.0E-89 | 1.5 | 0.82 | 0 |
| | SD | Short | PC1 | 0.31 | 2.0E-98 | 0.19 | 3.0E-36 | 0.40 | 2.0E-162 | 0.08 | 8.0E-07 | 0.19 | 6.0E-35 | 0.46 | 7.0E-78 | 0.8 | 0.65 | 0 |
| | SD | Short | PC2 | 0.37 | 4.0E-143 | 0.24 | 5.0E-60 | 0.47 | 3.0E-239 | 0.10 | 2.0E-11 | 0.20 | 2.0E-38 | 0.58 | 2.0E-73 | 0.9 | 0.68 | 0 |

For relationships with ALSFRS-R and right-left agreement, p-values for Pearson's correlation were computed using a Student's t distribution for a transformation of the correlation (two-tailed test). Mann–Whitney U test was used for ALS versus control comparisons (two-sided test) and Cohen's d was used to measure effect size. The Benjamini–Hochberg method was used to adjust for multiple comparisons and corrected p-values are reported.

*Stat* statistic, *Dur* duration, *Dir* direction, *Dist* distance, *Vel* velocity, *Accel* acceleration, *SM* submovement, *M* mean, *SD* standard deviation, *AI* activity intensity, *PC* principal component, *ALSFRS-R ALS* Functional Rating Scale-Revised, *ICC* intraclass correlation coefficient, *r* Pearson correlation coefficient, *es* effect size.

progressed faster over time than ALSFRS-R total (−0.73 ± 0.74 SD/year, $p = 0.007$ and $p = 0.017$, respectively; Fig. 4C). Female and male participants had nearly identical pairwise model progression rates (−0.86 ± 0.69 SD/year and −0.87 ± 0.70, respectively). Hypothetical clinical trial sample size estimates were smallest for the pairwise model ($N = 76$), followed by the regression model ($N = 86$), and ALSFRS-R ($N = 121$). Pairwise model and the regression model scores did not progress for control participants and were significantly different between ALS and control participants ($p = 0.0004$ and $p = 0.0006$, respectively; Fig. 4C). When using the mean rate of change of all four limbs, the pairwise model and ALSFRS-R total score were not significantly different (−0.56 ± 0.51 SD/year versus −0.73 ± 0.74 SD/year, $p = 0.12$) and ALSFRS-R total score was more sensitive than the regression model (−0.73 ± 0.74 SD/year versus −0.54 ± 0.56 SD/year, $p = 0.037$). Hypothetical clinical trial sample size estimates were again smallest for the pairwise model ($N = 101$), due to lower population variance in rate of change, followed by ALSFRS-R ($N = 121$), and the regression model ($N = 126$).

## Discussion

We have shown that data from inexpensive sensors worn on limbs at home during natural behavior can produce reliable, sensitive, and interpretable measures of gross and fine motor function in individuals with ALS. Ankle movement features derived from accelerometer data were highly consistent across right and left ankles and were in agreement with gross motor function as assessed on ALSFRS-R, both in terms of cross-sectional severity and in terms of rate of change over

time. Similarly, wrist movement features were highly consistent across right and left wrists and were in agreement with fine motor function on ALSFRS-R. Although there was strong right-left limb agreement at a population level, arm-leg agreement showed only moderate agreement, and some individuals were observed to have different rates of progression for each limb. Taking the score of the limb with the maximum progression rate produced a motor outcome measure that was consistent with but more sensitive than the current primary outcome measure in most ALS trials (ALSFRS-R), resulting in smaller hypothetical clinical trial sample size estimates.

The analysis approach for quantifying motor function in ALS centered on the extraction and characterization of motor primitives called submovements during natural behavior, which was previously developed for quantifying motor function in ataxia-telangiectasia[20] and adult cerebellar ataxias[21]. There is evidence that motor control is achieved by combining submovements to compose complex voluntary motor behaviors[24–27] and that submovements change in a consistent manner with the state of the motor system. In various contexts, such as infant development[28], aging[29], stroke recovery[30], and ataxia[31,32], submovements extracted from specific motor tasks reflect changes in motor function. During natural at-home behavior, ankle submovement distance, peak velocity, and peak acceleration are smaller in adults with spinocerebellar ataxias and multiple system atrophy compared to controls and become progressively smaller and less variable as self-reported function decreases and ataxia severity increases[21]. The submovement analysis approach contrasts with a prior analysis of task-

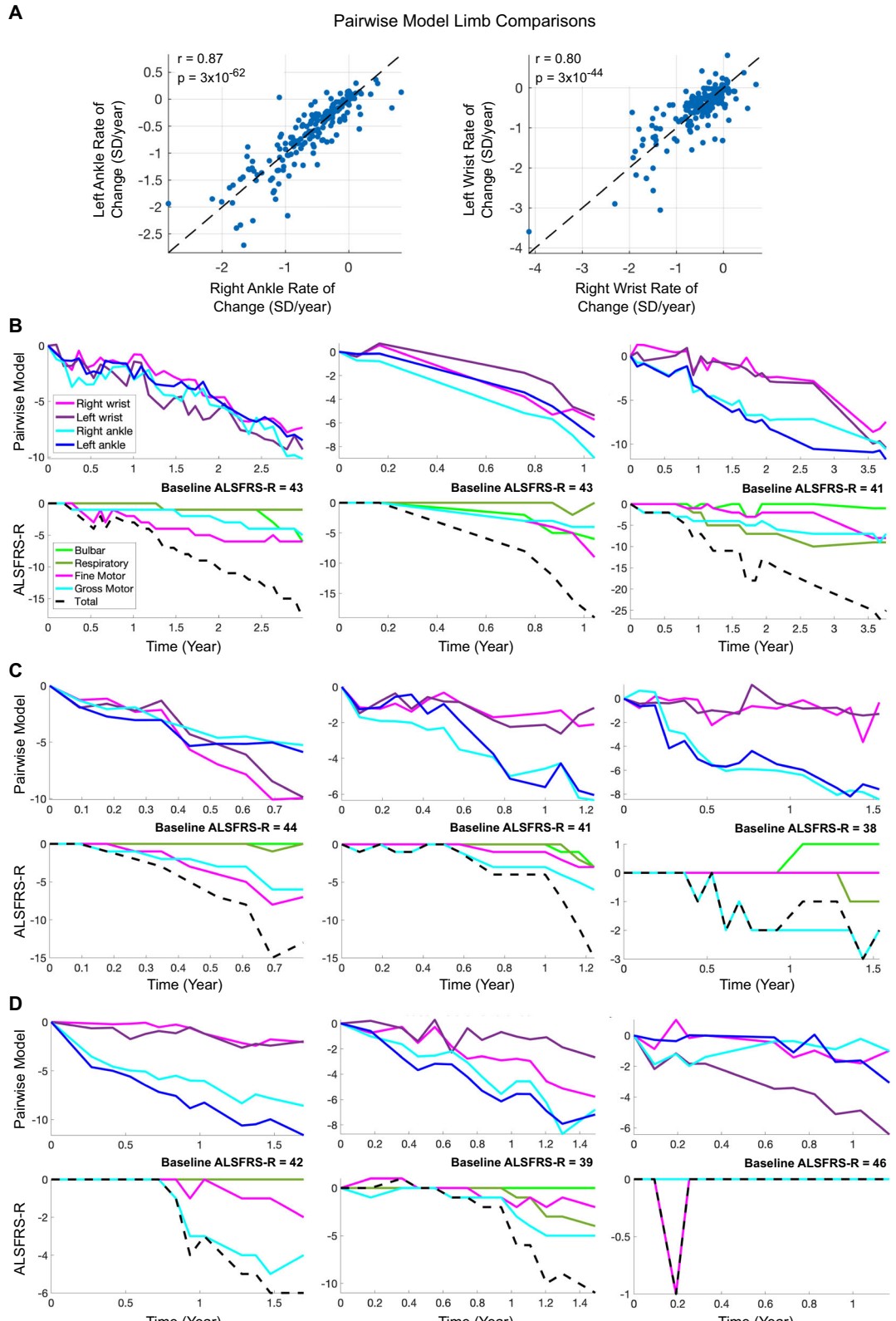

**Fig. 3 | Longitudinal data from each limb. A** Agreement in rate of change of the pairwise model score between right and left ankles, and right and left wrists. P-values for Pearson's correlation were computed using a Student's t distribution for a transformation of the correlation (two-tailed test). Each point represents an individual with ALS. Source data are provided as a Source Data file.
**B–D** Longitudinal trajectories for nine individuals with ALS, with sensor-based pairwise model scores for each limb shown in the top panel and ALSFRS-R scores shown in the bottom panel. Individuals were observed to have similar trajectories for all four limbs (**B**), similar trajectories for both ankles and both wrists (**C**), or divergent trajectories for one or more limbs (**D**). SD standard deviation, ALSFRS-R ALS Functional Rating Scale-Revised.

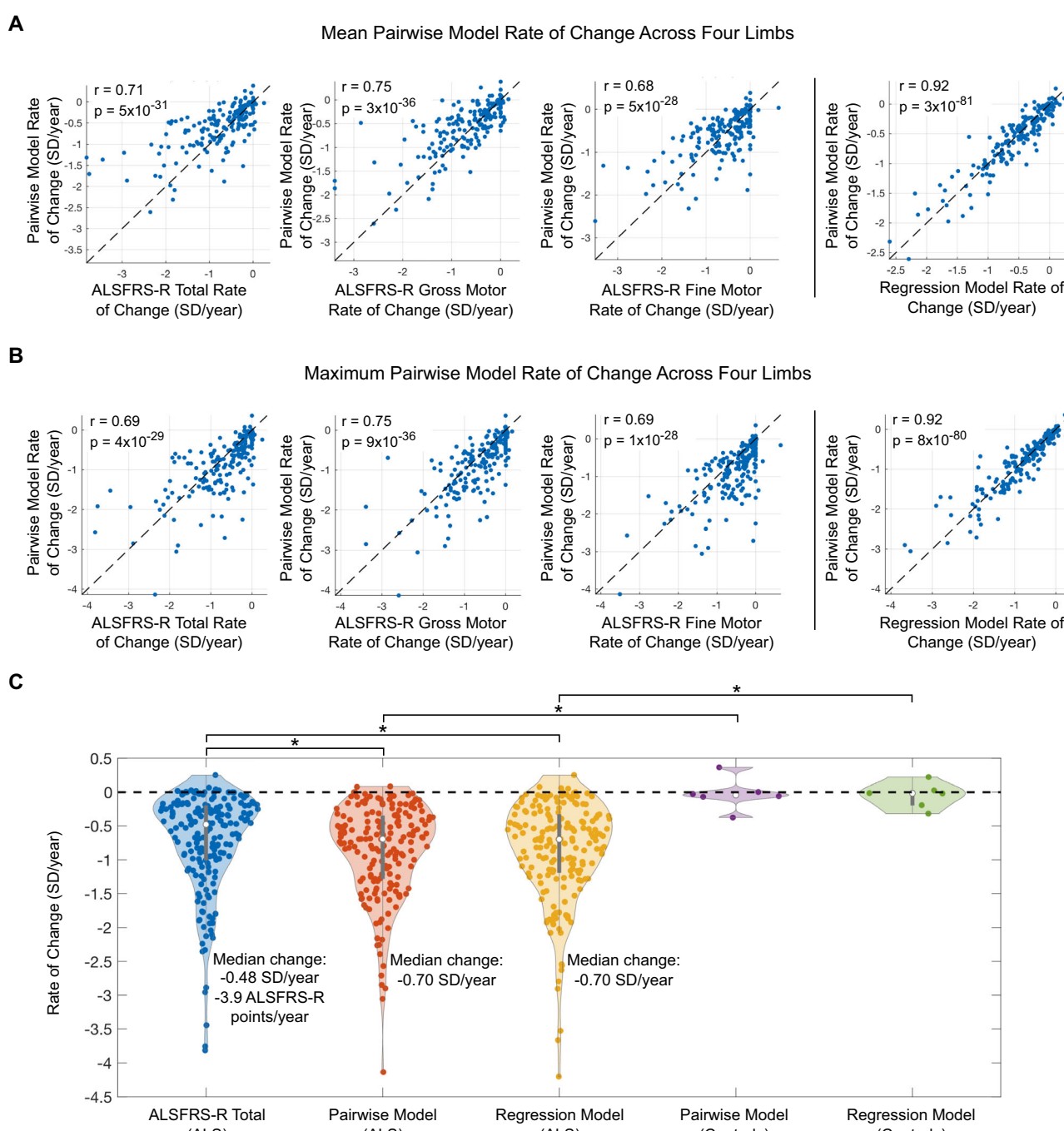

**Fig. 4 | Rate of change comparison between sensor-based models and ALSFRS-R. A** Mean pairwise model rate of change across four limbs compared with ALSFRS-R total score, gross motor subscore, and fine motor subscore (left) and compared with the mean regression model rate of change (right). **B** Maximum pairwise model rate of change across four limbs compared with ALSFRS-R total score, gross motor subscore, and fine motor subscore (left) and compared with the maximum regression model rate of change (right). *P*-values for Pearson's correlation were computed using a Student's t distribution for a transformation of the correlation (two-tailed test). **C** Violin plot comparing the distributions of ALSFRS-R rate of change, pairwise model rate of change (using fastest progressing limb), and regression model rate of change (using fastest progressing limb). Center point is median; gray line indicates the interquartile range ($n = 188$ ALS participants and $n = 6$ control participants). Rates of change were compared with '*' indicating a statistically significant comparison ($p < 0.05$, two-sided Mann–Whitney *U* test): $p = 0.007$ for pairwise model versus ALSFRS-R (ALS participants), $p = 0.017$ for regression model versus ALSFRS-R (ALS participants), $p = 0.0004$ for ALS versus control participants (pairwise model), and $p = 0.0006$ for ALS versus control participants (regression model). For **A**–**C**, each point represents a participant. SD standard deviation, ALSFRS-R ALS Functional Rating Scale-Revised. Source data are provided as a Source Data file.

free, at-home measurement in 42 individuals with ALS using waist-worn accelerometers, which quantified overall activity levels (e.g., activity count, percent of day active)[19]. Although overall motor activity is a pertinent outcome in ALS, it is reliant on full-day sensor wear and is likely more susceptible to day-to-day changes in behavioral context (e.g., travel, systemic illness, sleep quality), requiring careful consideration of reliability.

Based on our literature review, limb submovement features have not been previously studied in ALS. Several studies, however, have investigated the relationship between muscle strength (a direct cause

of motor impairment in ALS[33]) and submovement characteristics. In a heterogeneous population of individuals with motor impairments (e.g., spinal cord injury, cerebral palsy, stroke), participants were asked to perform a computer-based pointing task and a mechanical dynamometer was used to measure grip strength and pinch strength[34]. The authors found that the number of submovements per pointing movement was negatively correlated with grip strength (the movement was composed of smaller submovements as grip strength decreased) and that the velocity of movement was directly proportional to grip strength[34]. In another study of individuals with hemiparesis secondary to stroke, it was found that peak arm reaching velocity was influenced most by shoulder, elbow, and wrist flexor and extensor muscle strength (58% of variance explained), measured using a hand-held dynamometer[35]. In a study of individuals without motor disability, submovement organization was examined as participants tracked a small or large dot on a screen with a pen placed on a digitizer tablet, while simultaneously recording activity from muscles in the neck and upper extremity using surface electrodes[36]. When tracking the smaller target, extensor and flexor muscles of the forearm activated more strongly, and submovements were found to have increased peak velocities[36].

These studies support that there is a robust relationship between muscle strength and submovement features, in particular peak velocity. Consistent with these studies, we found that wrist and ankle submovements from individuals with ALS had smaller velocities, accelerations, and distances traversed. Submovement peak velocity was the only highly selected feature in both the right ankle and the right wrist pairwise models, demonstrating its importance for measuring disease progression ALS. This supports a model in which muscle weakness and decreased muscle activation caused by motor neuron pathology gives rise to slower and smaller submovements during everyday limb movement. Further supporting this model, are the parallels in left-right symmetry observed in the present study with the left-right symmetry observed in large studies of hand-held dynamometry (HHD)[23] and Accurate Test of Limb Isometric Strength (ATLIS)[37] in ALS. Individual arm and leg muscles were found to correlate strongly with the identical muscles on the contralateral side, both in terms of cross-sectional strength measurements ($r = 0.65-0.90$) and also in terms of rate of change over time ($r = 0.43-0.82$)[23]. We observed similar side-to-side cross-sectional and rate of change symmetry in individual submovement features (cross-sectional $r = 0.65-0.97$) and composite models (cross-sectional $r = 0.82-0.95$; rate of change $r = 0.80-0.87$). The high degree of correlation between right and left limbs and the observation that handedness and footedness can change over time in individuals with ALS, motivated our designation of limbs as right and left rather than dominant and nondominant. Interestingly, side-to-side cross-sectional symmetry of the leg was stronger than side-to-side symmetry of the arm here and in the HHD study. This may have implications for how ALS disease pathology spreads and highlights a potential future application of this technology in characterizing phenotypic spread across limbs in a continuous and granular fashion, for example in presymptomatic gene carriers. This also supports that submovement characteristics may be a suitable proxy for muscle strength in ALS, and offers an advantage over HHD and ATLIS of being able to measure strength continuously over multiple days, during the individual's own selection of behaviors, and without relying on participant effort or evaluator training and strength. Thus, it may produce more reliable, ecologically valid, and scalable measures of muscle strength and motor function. It may also apply to other neurological conditions that affect muscle strength. A future study that collects HHD and/or ATLIS measurements along with submovements from accelerometer data would help clarify the relationship between strength and submovements in ALS.

As discussed above, strong side-to-side correlations of ankle and wrist submovement features and composite models were observed.

This is consistent with previously reported strength measurements in ALS[23,37], but also highlights the robustness of the submovement measures that are generated independently from each limb's movement during natural behavior at home. Ankle submovement measures correlated strongly with ALSFRS-R gross motor subscore (both cross-sectional scores and rate of change) and wrist submovement measures correlated strongly with ALSFRS-R fine motor subscores. We found high test-retest reliability of the sensor-based features and composite models. Finally, two machine learning models trained based on different information (pairwise model trained on longitudinal change; regression model trained on ALSFRS-R) generated composite scores that had strong agreement in the rate of disease progression ($r = 0.92$). These properties support that sensor-derived submovements obtained during natural behavior provide highly robust measures of disease severity for each limb. Since each limb can be reliably and independently measured, these data support the use of the fastest progressing limb's rate of progression in order to obtain a personalized overall measure that is more sensitive for measuring disease change than ALSFRS-R and which may be more responsive to therapeutic intervention. However, the choice of if and how to combine severity measures from each limb can be determined based on the clinical application as well as on the individual's prior clinical trajectory (for example in a run-in period prior to intervention in a clinical trial). To achieve maximal sensitivity for disease change, these data support collecting movement information from all four limbs. Given the high reliability of the sensor-based measures and since each limb is analyzed independently, it is not necessary to wear all four sensors simultaneously. An alternative design could be to wear one sensor at a time and rotate its location on the body in one-week intervals. Thus, each limb is still measured continuously for one week each month.

Two different supervised machine-learning approaches were used to create composite measures of overall motor impairment for each limb based on the collection of sensor-based movement features. One used the traditional approach of training a regression model to predict severity as measured by ALSFRS-R. The other approach learned the trajectory of disease progression (in feature space) from the longitudinal data and computed how far the individual had moved along that trajectory without ever having access to rating scale data (i.e., pairwise model). Despite the very different training approaches, both models were highly consistent in their estimates of progression rate ($r = 0.92$) and were similarly consistent with ALSFRS-R total's progression rate ($r = 0.69$ and $r = 0.71$). The pairwise model was highlighted in analysis for three main reasons: (1) it had higher reliability than the regression models, (2) the consistency with ALSFRS-R in cross-section and in the rate of change was striking given that it had no chance to "overfit" to the clinical score, and (3) the pairwise modeling approach may be useful for other diseases where the existing clinical rating scale is less sensitive for capturing disease change. Furthermore, the pairwise modeling approach can be extended in a number of ways, for example by filtering comparisons, changing the type of classifier used, and aggregating data across multiple disease populations.

The large and longitudinal dataset generated by the ALS TDI Precision Medicine Program, consisting of 376 individuals with ALS who wore four sensors for multiple hours and days at home and with 188 participants who wore the four sensors longitudinally over a minimum of 0.75 years (median of 15 times over 1.5 years), supports the feasibility of the at-home passive data collection approach from both a patient and clinical operations perspective. Notably, although the Actigraph GT3X device was used in the current study, different devices were used in prior studies in ataxias[20,21], and the analytic approach presented here can likely be applied to any wearable sensor that captures triaxial accelerometer data at a minimum of 30 Hz, including consumer-grade sensors.

There were some limitations to the study. There was a relatively small number of controls included in the study and the controls were

not age matched. However, the size and characteristics of the control sample do not affect the main conclusions of the study. There was heterogeneity in the number of hours each participant wore the sensors at home. This was mitigated in part by only including days in which sensors were worn for at least 3 h. We anticipate higher reliability estimates of all sensor-based measures if participants are explicitly asked to wear the sensor throughout the entire day with exception of bathing (and night if possible). Finally, as expected, the severity estimates based on limb movement did not correlate well with bulbar and respiratory function. These functions are represented in ALSFRS-R and other digital strategies (e.g., video-based analysis of facial movement or speech analysis[17,38–40]) are needed to quantify these important motor domains in ALS.

In summary, we have shown that a submovement-based analysis of natural behavior at home using wearable sensors produces interpretable, reliable, sensitive, and ecologically valid measures of gross and fine motor function in ALS. This technology has properties that support its use as an outcome measure in ALS clinical trials with the potential to reduce the cost and size of future trials. The use of inexpensive sensors, worn at home with minimal instruction and no eligibility limitations, could increase access to clinical trials and support virtual clinical trials in ALS. It may also support the routine clinical care of individuals with ALS by providing clinicians and patients with an objective and reliable motor assessment that can be passively obtained at home with a relatively low burden and cost.

## Methods

This research study was conducted in accordance with the ethical principles posited in the Declaration of Helsinki - Ethical Principles for Medical Research Involving Human Subjects. Protocol approval was provided by the institutional review board (ADVARRA CIRBI). Every participant consented to participate in this research by signing an IRB approved informed consent form. There was no participant compensation in this study. Gender of participants was determined based on self-report and was not explicitly considered in the study design.

### Wearable sensor data processing and feature types

Continuous triaxial accelerometer data collected at 30 Hz was obtained from Actigraph GT3X devices (one for each limb). The cost of a single sensor ranged from $234-433 over the course of the study. Participants received a different sensor at each time point in the study. Any repeat use of a device by a participant would have been coincidental. In prior work, each participant's wearable sensor data were manually partitioned into day and night segments based on changes in each participant's daily activity level represented in the accelerometer data[20,21,41]. However, given the large size of this dataset, day segments were automatically partitioned to include data collected between 7:21 am and 11:27 pm, the pooled mean estimates of sleep offset and sleep onset in the oldest age group (15–18 year old's) studied in Galland et al.[42], while accounting for each individual's time zone. Visual inspection of random samples of 24-h periods of accelerometer data from multiple participants demonstrated that these times produced reasonable day-night segmentations. Data analysis focused on daytime segments. Gravity and high-frequency noise were removed from the acceleration time series using a sixth-order Butterworth filter with cutoff frequencies of 0.1 and 20 Hz[20,21,41,43].

Several classes of features were extracted from daytime ankle and wrist sensor data as in prior work[20,21]. These included *total power* in the 0.1-5 Hz frequency range and features based on the distribution of *activity intensity* computed in 1-second time bins. Features were also extracted from "activity bouts" and from submovements. Supplementary Table 1 provides a description of the 85 features extracted from ankle and wrist sensor data. Based on prior work, single feature

analysis was performed on a subset of 24 submovement features of interest as described in the main text.

### Severity estimation models

Supervised machine learning approaches were used to create composite severity scores that aggregate over the 85 movement features. Separate models were trained for each limb. The pairwise comparison approach is described in Fig. 2 and the main text. To ensure that the pairwise model did not inadvertently learn longitudinal changes resulting from changes in device settings, comparisons were only allowed between sessions that had the same critical firmware version (where raw data were collected in an identical way). Five-fold cross-validation was used: for each fold comparisons from 80% of ALS participants were used to train a classification model and the model weights were applied to data from the held-out 20% of participants to generate severity scores for each session. Additionally, we trained linear regression models with L1 regularization (i.e., lasso regression)[22] to predict ALSFRS-R total, ALSFRS-R gross motor subscore (ankle sensor data only), and ALSFRS-R fine motor subscore (wrist sensor data only). Five-fold cross-validation was also used to evaluate the performance of the regression models. For both the pairwise models and the regression models, each feature was z-score transformed prior to model training such that feature value ranges and model weights were comparable. Pearson correlation coefficient was used to measure performance, with each model compared with ALSFRS-R.

### Statistical analyses

Statistical analyses were completed in MATLAB version R2022a (Mathworks, Natick, MA). In longitudinal data analysis, each participant's progression rate for a given measure was determined by fitting a linear regression model to the individual's longitudinal data for the measure and using the slope of the curve to represent a progression over time[23]. The mean and standard deviation of the slope for each measure were computed across all ALS participants. For hypothetical clinical trial sample size estimates, we used a one-sample model for a continuous outcome[44] as described in Rutkove et al.[16] with the same model parameters: 90% power to detect a 30% mean change in progression rate, with two-sided $P$ values and a significance level of 0.05.

The non-parametric Mann–Whitney $U$ test was used to determine individual feature differences between disease and control groups and Cohen's $d$ was used to measure effect size. The Mann–Whitney $U$ test was also used to determine differences in the rate of change between different assessments. The Benjamini–Hochberg method was used to adjust for multiple comparisons and corrected $p$-values are reported[45]. Corrected $p$-values <0.05 were considered significant. Single-measure intraclass correlation coefficients (ICCs) were used to determine the test–retest reliability of features and composite scores. To evaluate the reliability of sensor-based features, features were computed from data recorded in first half of the days in the session (e.g., days 1–4) and the second half of the days in the session (e.g., days 5–8), separately, and ICCs were computed using a 2-way mixed effects model[46]. Pearson correlation coefficients and $p$-values were used to evaluate the relationship between sensor-based features and ALSFRS-R. As above, the Benjamini–Hochberg method was used to adjust for multiple comparisons[45].

### Reporting summary

Further information on research design is available in the Nature Portfolio Reporting Summary linked to this article.

### Data availability

The GTX3 accelerometer data and associated ALSFRS-R data are available upon request because file sizes necessitate coordinated data

transfer. Access can be obtained by visiting https://www.als.net/arc/data-commons/ and requesting the dataset by submitting accession code 06162023. Source data are provided with this paper.

## Code availability

The code to train the pairwise model is available in the github repository https://github.com/neuropheno-org/Pairwise_model_code.

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

## Acknowledgements

The authors thank James Berry and Katherine Burke for helpful discussions. We also thank the community of people with ALS who contributed data to these studies. The study was supported in part by NIH R01 NS117826. (A.S.G.).

## Author contributions

F.V. conceived of the translational research program that resulted in the data collected and analyzed in this manuscript. A.S.G., S.P., A.P., and F.V. conceived of the study objectives. A.P. and F.V. contributed to data collection efforts. S.P. performed ingestion and preprocessing of the dataset. A.S.G. performed analysis of the dataset. A.S.G., S.P., A.P., and F.V. contributed to the interpretation of the results. A.S.G. took the lead in writing the manuscript. All authors provided critical feedback and helped shape the research, analysis, and manuscript.

## Competing interests

For the methods for extracting and characterizing submovements from wearable sensor data, a PCT (US2022/081374) was filed on December 12, 2022, titled "System and method for clinical disorder assessment". An earlier US Provisional Application (Serial No. 63/288,619) was filed on December 12, 2021. The patent applicant is the institution (Massachusetts General Hospital) and inventor is Anoopum Gupta. The remaining authors declare no competing interests.
