## [Peer Review File · Nature Communications]

At-home wearables and machine learning sensitively capture disease progression in amyotrophic lateral sclerosisREVIEWER COMMENTS

Reviewer #1 (Remarks to the Author):

The authors present a large study of longitudinal accelerometer data comprising 376 patients with ALS. The authors attempt to identify new metrics of disease progression that could be used as alternative to current gold standard outcomes like the ALS functional rating scale (ALSFRS-R). The authors illustrate how some metrics may be more sensitive to change over time than survey-based outcomes. The strength of the study is the large sample size and longitudinal design; the weaknesses are the limited information on adherence & feasibility, and potential limitations in the statistical analysis. I have the following remarks:

1. The introduction provides an accurate overview of the current state of art, what is known and how the author's study adds to the literature. I appreciate the differentiation between passive and active tasks, and their focus on passive tasks as this indeed seems the best way to avoid current limitations in other outcomes (ALSFRS-R). The aim and objective of the study are well described.

2. I appreciate the "pairwise model" approach, disregarding the clinical outcome and using only the longitudinal wearable data to identify those features that show the largest change over time. This is a major strength of the paper as an imperfect prediction of the ALSFRS-R is actually desired: one wants to improve on what the ALSFRS-R measures and its sensitivity. The authors focus on the rate of change, but the variability in rates between patients plays also a significant role. These two factors determine the 'sensitivity' of a longitudinal marker ('signal to noise'); could the authors provide insight in the variability between patients relative to the average rate of decline?

3. Results: the monitoring protocol is quite extensive with 4 accelerometers per patient. Could the authors provide some insight in the adherence of their monitoring protocol? The protocol seems to dictate a 7-day wear-time period from 7:21 am to 11:27 pm; how many patients managed to complete this period (or 80% of this period?). In addition, some metrics of patient burden and retention would be insightful, e.g., the number of patients that withdrew their consent.

4. Could the author clarify the term "submovement properties" earlier in the manuscript, the explanation comes now after the results.

5. The ICC is based on the "data recorded in first half of the days in the session and the second half of the days". This is likely to give a systematic difference as patients may be more active in the first half compared to the second half due to fatigue. A more accurate comparison may have been using day 1 versus day 2; this would also provide better evidence for this statement in the discussion when comparing features with earlier reports: "..., it is reliant on full day sensor wear and is likely more susceptible to day-to-day changes in behavioral context".

6. Suggest to reword this: "..., each score was z-scored.." to each score was standardized (subtracting the mean divided by the standard deviation) and expressed as z-score. This should be clarified: "z-score/standard deviations", as the z-score is already a unit divided by the standard deviation, the standard deviation of the z-score is 1. I believe the author mean to say here: z-score/standard deviation of the rate of decline?

7. From the methods it is unclear how the rate of decline was calculated. There are no longitudinal models reported. From the data presented it looks like that per patient a regression model was fitted, and the regression coefficient/slope was extracted. This information should be provided as the statistical analysis cannot be replicated at the moment.

8. The pairwise comparison model is interesting (figure 2), but the description is lacking clarity and details. This makes it hard for the less statistical reader to digest what the authors have done.

Suppose one has 3 time points, there are 3 unique pairs (T1 vs 2, T1 vs 3, T2 vs 3), resulting the formula $n * (n-1) / 2$. From panel B these three comparisons would all receive a 0 as outcome (as T1 comes before T2, etc). To get also cases with a 1, however, one also needs to take the inverse comparisons (i.e. T2 vs 1, T3 vs 1, T3 vs 2). Thus, in this case, the number of comparisons used in the logistic regression model would be 6 rows of data, not 3 as the formula of # comparisons seems to suggest? Second, by calculating the delta for each feature between T1 – T2, T1 – T3, etc one assumes that the time difference between T1-T2, T1-T3 is equal. Clearly this is not the case, shouldn't the deltas not be divided by the delta time as well?

9. The results section could be written more concise, a lot of correlations and numbers are provided (most of which are already given in the table/figures). This dilutes a bit the main message. Table 1 is chaotic, what need the reader to see and learn from this? Second, I am unsure what the control data adds to this manuscript (e.g. Figure 4C?)

10. Using the "the limb with the fastest progression rate" has some complexities. This can of course be done after all that are collected, and could be OK for simple observation studies. For clinical trials however the outcomes and their analyses need to be prespecified prior to data look (else this could result in increased type 1 error).

11. General comment on discussion: this is quite lengthy and at some place too hypothetical and far away from the main study findings (e.g. the parts about submovement properties). It could be written more concise.

Reviewer #2 (Remarks to the Author):

Via longitudinal data collection over a time period of up to 7.5 years, this study describes a novel digital endpoint to assess ALS disease severity, finding the biomarker to be more sensitive to disease progression than current clinical scoring systems. The authorship team should be congratulated on a phenomenal study and manuscript detailing the use of multi-site triaxial accelerometers for the generation of a novel disease progression score for tracking functional capacity in patients with ALS. The study protocol is robust and well-executed, the data analysis is novel for the clinical field, and the manuscript is clear and thorough. I have added minor comments below to clarify a few points for the reader. Otherwise, this is an exceptional study and will be extremely valuable to the field going forward.

Comments and suggestions:

- In the title and throughout the manuscript, the authors refer to the Actigraph accelerometer as a "consumer-grade" wearable. While I believe the differentiation that the authors are drawing is to distinguish these from medical-grade devices, most researchers in the wearable community would refer to this device as a "research-grade" device (as it is made to collect raw accelerometry data) and only think of Fitbit/Apple Watch-type devices as "consumer-grade" devices. I would suggest avoiding the term "consumer-grade" in this manuscript.
- What was the status/progression of the disease when the participants were recruited?
- Within the methodology, please state whether the participant received the same sensors (serial numbers) for each data collection and site, or whether they received a new/different sensor for each session.
- Please include information about levels of wear time and study compliance over time and at each site (ie. did patients wear the wrist more than the ankle, did participants' compliance increase/decrease over time enrolled in the study)
- 24 hours seems to be a short period of time as a threshold for a valid session compared to most physical activity studies. I understand why you've used this, as you're not reporting overall activity levels. Please give slightly more explanation/defence for the 24-hour required wear time.

- It would be helpful to have a bit more explanation in the text on the nature of the composite severity score. (Is this created as a 0-1 score? Does the score go up/down with severity over time, etc.)
- With the "Longitudinal properties of ankle and wrist sensor data" section, you state, "and the right wrist pairwise model rate of change showed stronger agreement with...". Can you explain how you dealt with dominant/non-dominant wrists? It seems in this instance it would make sense to look at the dominant wrist and not the right wrist. It would help to get a bit of clarification in this area.
- In the discussion you state, "This was mitigated in part by only including days in which sensors were worn for at least 3 hours." – this is new information that I didn't find in the methods. Please add to the methods when describing valid data.
- A few times in the manuscript you mention "inexpensive" sensors. Can you comment on the approximate cost of these sensors?
- It is unclear from the methods why data is partitioned between 7:21 am and 11:27 pm. Following the reference, I still was unable to understand exactly where these times come from. Can you add a bit more context as to what is happening here and why these times exactly?
- From the perspective of the reader who might want to implement this methodology into a clinical trial, it would be helpful to get the authors' perspective on practical clinical implementation. Namely, does the data imply that we really need to place a sensor on all four limbs, or is there a specific site that one sensor would do? Perhaps this will be dictated by the goals of a future study in terms of gross/fine motor evaluation. In other words, how practical is this to implement and does it need to be exactly replicated, or are there sites/time periods that might be most efficient?

Reviewer #3 (Remarks to the Author):

In this manuscript, participants with ALS were asked to wear 4 commercially available movement sensors on each limb for 1 week per month. A host of movement measures were evaluated, and a model generated to summarize movements longitudinally over time. A longitudinal model of disease progression was generated, both from internal movement data and from comparison to a functional rating scale, the ALSFRS-R. Interesting and reasonable patterns were noted, including high correlation left versus right sides, higher correlations between change in upper extremity movement to the fine motor domain of the ALSFRS-R than the gross motor domain, with the reverse being true in the legs. The model generated progression rates that appear somewhat more rapid than the ALSFRS-R, and a measure scaled by variability also showed attractive properties.

This is a well done study, modeled closely to other studies which have been done on patients with spino cerebellar ataxias. The results are provocative and potentially important in ALS. I have several suggestions. First, movement sensors worn for many hours generate a huge dataset, and specific attributes were identified for further analysis. This report focuses on movements called submovements, based on previous ataxia studies. However, submovements are not well defined here; in fact, as a non-expert in this field, I could not find a clear definition in other ataxia papers. This paper would be significantly strengthened by a definition of submovements that made sense to the reader. Further, a description of how submovements were identified for analysis would be very beneficial.

A limitation of this study, and virtually all other natural history study, is that those who participate are usually not representative of participants in clinical trials. This has implications for what rate of progression is expected, and also limits to some extent any comparison of rate solely within this study. Therefore, more data on the characteristics of patients participating in this study are necessary; duration of symptoms, prestudy progression rate, bulbar vs limb onset, other demographic information is important to present.

With regard to analysis, submovement velocity seems to be the measure of choice. However, a clear comparison of the properties of this measure compared to the other measures assessed would be very

useful to present.

Finally, with regard to the comparison of submovement to ALSFRS-r, one meaningful way to express this is to do a power analysis based on a standard trial; eg, something like 6 mo, 25% change in slope, 80% power, or something similar.

Reviewer #1 (Remarks to the Author):

The authors present a large study of longitudinal accelerometer data comprising 376 patients with ALS. The authors attempt to identify new metrics of disease progression that could be used as alternative to current gold standard outcomes like the ALS functional rating scale (ALSFRRS-R). The authors illustrate how some metrics may be more sensitive to change over time than survey-based outcomes. The strength of the study is the large sample size and longitudinal design; the weaknesses are the limited information on adherence & feasibility, and potential limitations in the statistical analysis. I have the following remarks:

1. The introduction provides an accurate overview of the current state of art, what is known and how the author's study adds to the literature. I appreciate the differentiation between passive and active tasks, and their focus on passive tasks as this indeed seems the best way to avoid current limitations in other outcomes (ALSFRRS-R). The aim and objective of the study are well described.

Thank you very much for your feedback.

2. I appreciate the "pairwise model" approach, disregarding the clinical outcome and using only the longitudinal wearable data to identify those features that show the largest change over time. This is a major strength of the paper as an imperfect prediction of the ALSFRRS-R is actually desired: one wants to improve on what the ALSFRRS-R measures and its sensitivity. The authors focus on the rate of change, but the variability in rates between patients plays also a significant role. These two factors determine the 'sensitivity' of a longitudinal marker ('signal to noise'); could the authors provide insight in the variability between patients relative to the average rate of decline?

Thank you for this important comment. We have modified the manuscript (abstract and results) to report mean +/- standard deviation of the rate of decline (in place of median) and report hypothetical clinical trial sample size estimations. Slight wording adjustments were made to the abstract to fit within the 150 word limit.

Relevant sentence from Abstract:

"The approach produced scores that progressed faster than the gold standard ALS Functional Rating Scale-Revised (-0.86 +/- 0.70 SD/year versus -0.73 +/- 0.74 SD/year), resulting in smaller clinical trial sample size estimates (N=76 versus N=121)."

Relevant text from Results:

"Using the maximum rate of change, the pairwise model had a progression rate of -0.86 +/- 0.70 (mean +/- standard deviation) SD/year and the regression model had a progression rate of -0.86 +/- 0.74 SD/year. Both the pairwise and regression models progressed faster over time than ALSFRRS-R total (-0.73 +/- 0.74 SD/year, $p = 0.007$ and $p = 0.017$, respectively; Figure 4C). Female and male participants had nearly identical pairwise model progression rates (-0.86 +/- 0.69 SD/year and -0.87 +/- 0.70, respectively). Hypothetical clinical trial sample size estimates were smallest for the pairwise model (N=76), followed by the regression model (N=86), and ALSFRRS-R (N=121)."

3. Results: the monitoring protocol is quite extensive with 4 accelerometers per patient. Could the authors provide some insight in the adherence of their monitoring protocol? The protocol seems to dictate a 7-day wear-time period from 7:21 am to 11:27 pm; how many patients managed to complete this period (or 80% of this period?). In addition, some metrics of patient burden and retention would be insightful, e.g., the number of patients that withdrew their consent.

We have added language to further clarify the instructions participants received and how much time they wore the wrist and ankle devices. Participants were counseled to wear the devices as much as possible for a week, but a specific wear period was not prescribed. We have also included an analysis of patient burden and retention for the wearable sensors (please see added text below).

Relevant text from Results:

"Individuals were asked to wear a sensor on each wrist and ankle continuously for a week each month. Participants also performed a sequence of 5 limb-based exercises on alternating days, lasting a total of approximately 5 minutes. Participants were instructed that sensors must be worn during the brief exercises, but to also wear the sensors as much as possible throughout the week without further specifying periods of wear time."

AND

“Participants had a median of 9 days per session with at least 3 hours/day of data and averaged 8.9 hours/day of wear time for the wrist sensors and 7.4 hours/day for the ankle sensors. The duration of daily sensor wear time (averaged across the four sensors) decreased over the course of the study from an average of 9.1 hours/day (first session) to 6.5 hours/day (last session). To understand the burden of wearing the four sensors periodically over a 0.75-year period (relative to at-home self-report of ALSFRS-R), we identified the subset of the 402 individuals with adequate cross sectional data who did not wear the sensors for at least 0.75 years but continued to perform ALSFRS-R self-report for 90 days or more after the last time they wore sensors. This consisted of 39 participants or ~10% of the cohort who continued performing ALSFRS-R but stopped wearing the sensors.”

Relevant text from Discussion/Limitations:

“There was heterogeneity in the number of hours each participant wore the sensors at home. This was mitigated in part by only including days in which sensors were worn for at least 3 hours. We anticipate higher reliability estimates of all sensor-based measures if participants are explicitly asked to wear the sensor throughout the entire day with exception of bathing (and night if possible).”

4. Could the author clarify the term “submovement properties” earlier in the manuscript, the explanation comes now after the results.

We interchange submovement “properties” and submovement “features” throughout the manuscript. To improve clarity we now only use the term “feature”.

Relevant text from Results:

“Single feature analysis was performed on a subset of 24 key submovement (SM) features of interest. These included SM distance, peak velocity, and peak acceleration (8 features each). Mean and standard deviation were computed for short duration and long duration SMs in the primary and secondary directions of planar movement resulting in 8 features for each measurement type.”

5. The ICC is based on the “data recorded in first half of the days in the session and the second half of the days”. This is likely to give a systematic difference as patients may be more active in the first half compared to the second half due to fatigue. A more accurate comparison may have been using day 1 versus day 2; this would also provide better evidence for this statement in the discussion when comparing features with earlier reports: “..., it is reliant on full day sensor wear and is likely more susceptible to day-to-day changes in behavioral context”.

We have clarified that individual days are not split into the first and second halves of the day in order to compute ICC. Instead we take all days recorded in a session and split the data into data from the first set of days in the session (e.g., days 1-4) and the second set of days in the session (e.g., days 5-8) in order to compute ICCs.

Relevant text from Methods:

“To evaluate reliability of sensor-based features, features were computed from data recorded in first half of the days in the session (e.g., days 1-4) and the second half of the days in the session (e.g., days 5-8), separately, and ICCs were computed using a 2-way mixed effects model.”

6. Suggest to reword this: “.., each score was z-scored..” to each score was standardized (subtracting the mean divided by the standard deviation) and expressed as z-score. This should be clarified: “z-score/standard deviations”, as the z-score is already a unit divided by the standard deviation, the standard deviation of the z-score is 1. I believe the author mean to say here: z-score/standard deviation of the rate of decline?

Thank you, we agree that this describes the process more clearly and have updated the text as suggested. To clarify how we report rate of change for a score: after the score is standardized and expressed as a z-score, we model the score over time for an individual using linear regression with the slope of the line representing rate of change. Rate of change over time has units of z-score/year, which is equivalent to standard deviations(SD)/year. In the sentence above we used the symbol ‘/’ to indicate equivalence rather than division, which was a poor choice. We have reworded this below.

Relevant text from Results:

“The rate of change over time for each sensor-based composite score and ALSFRS-R score was modeled using linear regression, with the slope of the best fit line determining the rate of change²³. To compare the rate of change of

different scores, each with a different range of values, each score was standardized (subtracting the mean and dividing by the standard deviation) and expressed as a z-score. Rate of change for each score was reported as z-score per year or equivalently as standard deviations (SD) per year.”

7. From the methods it is unclear how the rate of decline was calculated. There are no longitudinal models reported. From the data presented it looks like that per patient a regression model was fitted, and the regression coefficient/slope was extracted. This information should be provided as the statistical analysis cannot be replicated at the moment.

Thank you for catching this omission in the Methods. Your assumption is correct. We have added the details in the Statistical Analysis section.

Relevant text from Methods:

“In longitudinal data analysis, each participant's progression rate for a given measure was determined by fitting a linear regression model to the individual's longitudinal data for the measure and using the slope of the curve to represent progression over time. The mean and standard deviation of the slope for each measure was computed across all ALS participants. For hypothetical clinical trial sample size estimates, we used a one sample model for a continuous outcome⁴⁴ as described in Rutkove et al¹⁶ with the same model parameters: 90% power to detect a 30% mean change in progression rate, with two-sided *P* values and a significance level of 0.05.”

8. The pairwise comparison model is interesting (figure 2), but the description is lacking clarity and details. This makes it hard for the less statistical reader to digest what the authors have done. Suppose one has 3 time points, there are 3 unique pairs (T1 vs 2, T1 vs 3, T2 vs 3), resulting the formula $n * (n-1) / 2$. From panel B these three comparisons would all receive a 0 as outcome (as T1 comes before T2, etc). To get also cases with a 1, however, one also needs to take the inverse comparisons (i.e. T2 vs 1, T3 vs 1, T3 vs 2). Thus, in this case, the number of comparisons used in the logistic regression model would be 6 rows of data, not 3 as the formula of # comparisons seems to suggest? Second, by calculating the delta for each feature between T1 – T2, T1 – T3, etc one assumes that the time difference between T1-T2, T1-T3 is equal. Clearly this is not the case, shouldn't the deltas not be divided by the delta time as well?

Thank you, we were missing a key detail in the description regarding this point. For each of the possible $n * (n-1) / 2$ pairwise comparisons, the order is randomized such that for ~50% of the comparisons the outcome is 0 (earlier time point minus latter time point) and for ~50% of comparisons the outcome is 1 (later time point minus earlier time point). We have added this information to the Figure caption that describes the model.

Relevant text from Figure 2 caption:

“**B**) The model takes two 85-dimensional feature vectors or samples (S_1 and S_2) from a single individual as input, representing that individual's motor function at two different points in time (t_m and t_n). For each comparison, the samples at times t_m and t_n are randomly assigned to be S_1 and S_2 , such that for approximately half the comparisons S_1 is the earlier time sample (as shown in B.1.) and for the other half S_1 is the later time sample (B.2.). The element-wise difference between the two vectors is computed ($S_1 - S_2$), representing the direction of change in feature space. This difference vector is the input to a binary classifier (logistic regression) which learns to predict whether the direction of change reflects disease progression (S_2 is temporally after S_1) or reflects disease improvement (S_2 is temporally before S_1).”

9. The results section could be written more concise, a lot of correlations and numbers are provided (most of which are already given in the table/figures). This dilutes a bit the main message. Table 1 is chaotic, what need the reader to see and learn from this? Second, I am unsure what the control data adds to this manuscript (e.g. Figure 4C?)

We have revised the results section to remove less important correlations/numbers from the text and make it more concise. We have also split Table 1 into two tables (one for the wrist and one for the ankle) and moved the model rows to the top of the table for emphasis. Regarding the control data in Figure 4C, although it is a small sample, we believe it's important to highlight that the control population did not show progression over time for the sensor-based models.

10. Using the “the limb with the fastest progression rate” has some complexities. This can of course be done after all that are collected, and could be OK for simple observation studies. For clinical trials however the outcomes and their analyses need to be prespecified prior to data look (else this could result in increased type 1 error).

We agree that the specific context of use will be important in determining the number of limbs to measure and how to combine information across limbs. In the revised manuscript, we show that taking the mean of the four limbs also reduces the sample size estimate for a hypothetical clinical trial. It may also be possible to identify the fastest progressing limb in a run-in period prior to intervention (using sensors or clinically). Finally, it still may be possible for an outcome measure for a trial to be the progression of the limb with maximal progression over the duration of the trial, without prior knowledge of the fastest progressing limb.

Relevant text in Discussion:

“However, the choice of if and how to combine severity measures from each limb can be determined based on the clinical application as well as on the individual’s prior clinical trajectory (for example in a run-in period prior to intervention in a clinical trial).”

11. General comment on discussion: this is quite lengthy and at some place too hypothetical and far away from the main study findings (e.g. the parts about submovement properties). It could be written more concise.

Thank you for this suggestion, we have condensed the paragraph discussing submovement properties in other populations and replaced that part with a single sentence.

Condensed sentence in Discussion:

“In various contexts, such as infant development²⁸, aging²⁹, stroke recovery³⁰, and ataxia^{31,32}, submovements extracted from specific motor tasks reflect changes in motor function.”

Reviewer #2 (Remarks to the Author):

Via longitudinal data collection over a time period of up to 7.5 years, this study describes a novel digital endpoint to assess ALS disease severity, finding the biomarker to be more sensitive to disease progression than current clinical scoring systems. The authorship team should be congratulated on a phenomenal study and manuscript detailing the use of multi-site triaxial accelerometers for the generation of a novel disease progression score for tracking functional capacity in patients with ALS. The study protocol is robust and well-executed, the data analysis is novel for the clinical field, and the manuscript is clear and thorough. I have added minor comments below to clarify a few points for the reader. Otherwise, this is an exceptional study and will be extremely valuable to the field going forward.

Thank you very much!

Comments and suggestions:

- In the title and throughout the manuscript, the authors refer to the Actigraph accelerometer as a “consumer-grade” wearable. While I believe the differentiation that the authors are drawing is to distinguish these from medical-grade devices, most researchers in the wearable community would refer to this device as a “research-grade” device (as it is made to collect raw accelerometry data) and only think of Fitbit/Apple Watch-type devices as “consumer-grade” devices. I would suggest avoiding the term “consumer-grade” in this manuscript.

Thank you for this point, we have removed consumer-grade throughout the manuscript and title as suggested. We wanted to draw a distinction between these devices and more expensive medical- and laboratory-grade systems, but we agree that a further distinction between the Actigraph device and smartwatches is necessary. We have added text to the discussion to indicate that the analytic approach can likely be applied to data from consumer-grade devices.

New title: “At-home wearables and machine learning sensitively capture disease progression in amyotrophic lateral sclerosis”

Relevant text in Discussion:

“Notably, although the Actigraph GT3X device was used in the current study, different devices were used in prior studies in ataxias^{20,21}, and the analytic approach presented here can likely be applied to any wearable sensor that captures triaxial accelerometer data at a minimum of 30 Hz, including consumer-grade sensors.”

- What was the status/progression of the disease when the participants were recruited?

We have provided this information in Figure 1 and have updated the text to highlight the information.

Relevant text in Results:

"Participant clinical and demographic data are shown in Figure 1A, including the median ALSFRS-R at study start and study end."

- Within the methodology, please state whether the participant received the same sensors (serial numbers) for each data collection and site, or whether they received a new/different sensor for each session.

We have added the following text to Methods:

"Participants received a different sensor at each time point in the study. Any repeat use of a device by a participant would have been coincidental."

- Please include information about levels of wear time and study compliance over time and at each site (ie. did patients wear the wrist more than the ankle, did participants' compliance increase/decrease over time enrolled in the study)

- 24 hours seems to be a short period of time as a threshold for a valid session compared to most physical activity studies. I understand why you've used this, as you're not reporting overall activity levels. Please give slightly more explanation/defence for the 24-hour required wear time.

Thank you for these suggestions, we have added the following text to the Results to address these two important points:

"Participants had a median of 9 days per session with at least 3 hours/day of data and averaged 8.9 hours/day of wear time for the wrist sensors and 7.4 hours/day for the ankle sensors. The duration of daily sensor wear time (averaged across the four sensors) decreased over the course of the study from an average of 9.1 hours/day (first session) to 6.5 hours/day (last session)."

AND

"The 24 hour session minimum for daytime data was chosen based on prior work demonstrating high reliability of daytime data across the first three and last three days in a week^{20,21}."

- It would be helpful to have a bit more explanation in the text on the nature of the composite severity score. (Is this created as a 0-1 score? Does the score go up/down with severity over time, etc.)

We have added these details for the pairwise model severity score in the Results section describing the model:

"The model produces a score in which lower values represent increased impairment (as in ALSFRS-R) and there is no lower or upper bound on the value of the score, although scores in the current population ranged from -11.3 to 9.6."

- With the "Longitudinal properties of ankle and wrist sensor data" section, you state, "and the right wrist pairwise model rate of change showed stronger agreement with...". Can you explain how you dealt with dominant/non-dominant wrists? It seems in this instance it would make sense to look at the dominant wrist and not the right wrist. It would help to get a bit of clarification in this area.

Thank you for raising this point. Given the high degree of correlation in rate of change between the right and left wrist pairwise models ($r = 0.8$, Figure 3A right panel) and due to the complication that hand dominance can change over time in some individuals with ALS, we chose to present data for the right wrist rather than dominant wrist.

Relevant text added to Discussion:

"The high degree of correlation between right and left limbs and the observation that handedness and footedness can change over time in individuals with ALS, motivated our designation of limbs as right and left rather than dominant and nondominant."

- In the discussion you state, "This was mitigated in part by only including days in which sensors were worn for at least 3 hours." – this is new information that I didn't find in the methods. Please add to the methods when describing valid data.

Thank you, this has been added to the section describing the filtering pipeline.

- A few times in the manuscript you mention “inexpensive” sensors. Can you comment on the approximate cost of these sensors?

We have added the cost of the sensors to the Methods section describing the sensor:

“The cost of a single sensor ranged from \$234-433 over the course of the study.”

- It is unclear from the methods why data is partitioned between 7:21 am and 11:27 pm. Following the reference, I still was unable to understand exactly where these times come from. Can you add a bit more context as to what is happening here and why these times exactly?

These precise times were used because they were the pooled mean estimates of sleep offset and sleep onset for the oldest age group reported in Table 4 of Galland et al.

The following text has been added/modified in the Methods:

“Given the large size of this dataset, day segments were automatically partitioned to include data collected between 7:21 am and 11:27 pm, the pooled mean estimates of sleep offset and sleep onset in the oldest age group (15-18 year old’s) studied in Galland et al,⁴³ while accounting for each individual’s time zone. Visual inspection of random samples of 24-hour periods of accelerometer data from multiple participants demonstrated that these times produced reasonable day-night segmentations.”

- From the perspective of the reader who might want to implement this methodology into a clinical trial, it would be helpful to get the authors’ perspective on practical clinical implementation. Namely, does the data imply that we really need to place a sensor on all four limbs, or is there a specific site that one sensor would do? Perhaps this will be dictated by the goals of a future study in terms of gross/fine motor evaluation. In other words, how practical is this to implement and does it need to be exactly replicated, or are there sites/time periods that might be most efficient?

Thank you for this suggestion. We have expanded our suggestions for practical implementation in the Discussion:

“Since each limb can be reliably and independently measured, these data support the use of the fastest progressing limb’s rate of progression in order to obtain a personalized overall measure that is more sensitive for measuring disease change than ALSFRS-R and which may be more responsive to therapeutic intervention. However, the choice of if and how to combine severity measures from each limb can be determined based on the clinical application as well as on the individual’s prior clinical trajectory (for example in a run-in period prior to intervention in a clinical trial). To achieve maximal sensitivity for disease change, these data support collecting movement information from all four limbs. Given the high reliability of the sensor-based measures and since each limb is analyzed independently, it is not necessary to wear all four sensors simultaneously. An alternative design could be to wear one sensor at a time and rotate its location on the body in one-week intervals. Thus, each limb is still measured continuously for one week each month.”

Reviewer #3 (Remarks to the Author):

In this manuscript, participants with ALS were asked to wear 4 commercially available movement sensors on each limb for 1 week per month. A host of movement measures were evaluated, and a model generated to summarize movements longitudinally over time. A longitudinal model of disease progression was generated, both from internal movement data and from comparison to a functional rating scale, the ALSFRS-R. Interesting and reasonable patterns were noted, including high correlation left versus right sides, higher correlations between change in upper extremity movement to the fine motor domain of the ALSFRS-R than the gross motor domain, with the reverse being true in the legs. The model generated progression rates that appear somewhat more rapid than the ALSFRS-R, and a measure scaled by variability also showed attractive properties.

This is a well done study, modeled closely to other studies which have been done on patients with spino cerebellar ataxias. The results are provocative and potentially important in ALS. I have several suggestions. First, movement sensors worn for many hours generate a huge dataset, and specific attributes were identified for further analysis. This report focuses on movements called submovements, based on previous ataxia studies. However, submovements are not well defined here; in fact, as a non-expert in this field, I could not find a clear definition in other ataxia papers. This

paper would be significantly strengthened by a definition of submovements that made sense to the reader. Further, a description of how submovements were identified for analysis would be very beneficial.

Thank you for this important point. We have added this description where we first introduce submovements in the Overview of the dataset.

Relevant text from Results:

“Submovement, activity bout, activity index, and spectral movement features (85 total) were extracted from each session as previously described^{20,21} (Figure 1C, Supplementary Table 1). Briefly, continuous triaxial accelerometer data was processed to identify activity bouts (short periods of continuous movement), which were projected onto a 2D plane using principal component analysis to identify the primary and secondary directions of motion^{20,21}. The acceleration time series was converted to a velocity time series via integration. Submovements (i.e., typically bell-shaped velocity-time curves flanked by zero velocity crossings, see Figure 1C) were then identified in the primary and secondary directions of motion and grouped into long and short duration submovements.”

A limitation of this study, and virtually all other natural history study, is that those who participate are usually not representative of participants in clinical trials. This has implications for what rate of progression is expected, and also limits to some extent any comparison of rate solely within this study. Therefore, more data on the characteristics of patients participating in this study are necessary; duration of symptoms, prestudy progression rate, bulbar vs limb onset, other demographic information is important to present.

Thank you for this suggestion. Additional clinical and demographic information have been provided, including duration of disease at start of study, bulbar vs limb onset, location, race/ethnicity, and family history of ALS.

Relevant text from Results:

“Participant clinical and demographic data are shown in Figure 1A, including the median ALSFRS-R at study start and study end. 95% of participants lived in the United States (41 states represented), 3% lived in Canada, and 1% lived in the United Kingdom. 93.5% of participants were White, 2% Hispanic, 2% Asian, 1% Black, <1% Middle Eastern, and <1% Polish. 15% of ALS participants had a family history of ALS.”

Updated clinical and demographic table:

	Cross Sectional Analysis		Longitudinal Analysis	
	ALS	Control	ALS	Control
Participants (N)	376	26	188	6
Female/Male (N)	129 / 247	12 / 14	62 / 126	3 / 3
Median Age at Study Start (Years)	57 (21-79)	33 (20-67)	58 (27-79)	36 (26-50)
Median Sessions Per Participant (N)	7 (1-74)	4 (1-71)	15 (4-74)	35 (11-71)
Median Study Time Span (Years)	0.8 (0-7.3)	0.4 (0-6.7)	1.5 (0.8-7.3)	5 (2.3-6.7)
Median Disease Duration at Study Start (Years)	1.9 (0-20.3)	N/A	2.1 (0-20.3)	N/A
Median ALSFRS-R Total at Study Start	41 (14-48)	48 (38-48)	41 (21-48)	48 (48-48)
Median ALSFRS-R Total at Study End	36 (5-48)	48 (38-48)	32 (5-48)	48 (48-48)
First Symptoms Include Upper Extremities (N)	159	N/A	81	N/A
First Symptoms Include Lower Extremities (N)	164	N/A	86	N/A
First Symptoms Include Bulbar Symptoms (N)	75	N/A	36	N/A
First Symptoms Include Respiratory Symptoms (N)	9	N/A	5	N/A

With regard to analysis, submovement velocity seems to be the measure of choice. However, a clear comparison of the properties of this measure compared to the other measures assessed would be very useful to present.

We have added text in the Results section to address this important point.

Relevant text from Results:

“Although SM peak velocity was strongly represented in the models, the properties of peak velocity at an individual feature level (e.g., relationships with ALSFRS-R, test-retest reliability) were comparable to SM acceleration and

distance features and showed weaker relationships with ALSFRS-R compared to the pairwise models (see Tables 1 and 2).”

Finally, with regard to the comparison of submovement to ALSFRS-r, one meaningful way to express this is to do a power analysis based on a standard trial; eg, something like 6 mo, 25% change in slope, 80% power, or something similar.

Thank you for this suggestion. We have now included a power analysis to compare the measures.

Relevant text from Results:

“Using the maximum rate of change, the pairwise model had a progression rate of -0.86 ± 0.70 (mean \pm standard deviation) SD/year and the regression model had a progression rate of -0.86 ± 0.74 SD/year. Both the pairwise and regression models progressed faster over time than ALSFRS-R total (-0.73 ± 0.74 SD/year, $p = 0.007$ and $p = 0.017$, respectively; Figure 4C). Female and male participants had nearly identical pairwise model progression rates (-0.86 ± 0.69 SD/year and -0.87 ± 0.70 , respectively). Hypothetical clinical trial sample size estimates were smallest for the pairwise model (N=76), followed by the regression model (N=86), and ALSFRS-R (N=121).”

Relevant text from Methods:
“For hypothetical clinical trial sample size estimates, we used a one sample model for a continuous outcome⁴⁴ as described in Rutkove et al¹⁶ with the same model parameters: 90% power to detect a 30% mean change in progression rate, with two-sided P values and a significance level of 0.05.”

REVIEWERS' COMMENTS

Reviewer #1 (Remarks to the Author):

The authors have addressed my comments sufficiently, thank you

Reviewer #2 (Remarks to the Author):

The authors should again be commended on a very well conducted study and manuscript. The authors have carefully and thoughtfully responded to each of my concerns. I have no further comments and recommend this manuscript for acceptance.

Reviewer #3 (Remarks to the Author):

The authors have been very responsive to my suggestions. I think the study is important and worthy of publication.

Response to Reviewers

Reviewer #1 (Remarks to the Author):

The authors have addressed my comments sufficiently, thank you

Reviewer #2 (Remarks to the Author):

The authors should again be commended on a very well conducted study and manuscript. The authors have carefully and thoughtfully responded to each of my concerns. I have no further comments and recommend this manuscript for acceptance.

Reviewer #3 (Remarks to the Author):

The authors have been very responsive to my suggestions. I think the study is important and worthy of publication.

Many thanks to the three reviewers for their insightful and constructive comments that have resulted in a stronger manuscript.